# Design of Dielectric Elastomer Actuator and Its Application in Flexible Gripper

**DOI:** 10.3390/mi16010107

**Published:** 2025-01-19

**Authors:** Xiaoyu Meng, Jiaqing Xie, Haoran Pang, Wenchao Wei, Jiping Niu, Mingqiang Zhu, Fang Gu, Xiaohuan Fan, Haiyan Fan

**Affiliations:** 1College of Mechanical and Electronic Engineering, Northwest A&F University, Yangling 712100, China; mxy0917@nwafu.edu.cn (X.M.); 18729071855@nwafu.edu.cn (H.P.); 2023056062@nwafu.edu.cn (W.W.);; 2College of Water Resources and Architectural Engineering, Northwest A&F University, Yangling 712100, China; 3Guangdong Association of Environmental Protecion Industry, Guangzhou 510045, China; 4Zhejiang Sunny Optical Company, Yuyao 315400, China

**Keywords:** dielectric elastomer actuator, soft gripper, pre-stretch, shape design, bending range

## Abstract

Dielectric elastomer actuators (DEAs) are difficult to apply to flexible grippers due to their small deformation range and low output force. Hence, a DEA with a large bending deformation range and output force was designed, and a corresponding flexible gripper was developed to realize the function of grasping objects of different shapes. The relationship between the pre-stretch ratio and DEA deformation degree was tested by experiments. Based on the performance test results of the dielectric elastomer (DE), the bending deformation process of DEAs with different shapes was simulated by Finite Element Method (FEM) simulation. DEAs with different shapes were prepared through laser cutting and the relationship between the voltage and the bending angle, and the output force of the DEAs was measured. The result shows that under uniaxial stretching, the deformation of the DEA in the stretching direction gradually increases and decreases in the unstretched direction with the increase in the pre-stretch ratio. Under biaxial stretching, DEA deformation increases with the increase in the pre-stretch ratio. The shape of the DEA has a certain influence on the bending deformation range under the same conditions, and the elliptical DEA has a larger bending deformation range and higher output force compared with the rectangular DEA and the trapezium DEA. The elliptical DEA can produce a bending deformation of 40° and an output force of 37.2 mN at a voltage of 24 kV. The three-finger flexible gripper composed of an elliptical DEA can realize the grasping of a paper cup.

## 1. Introduction

Benefiting from the capability to proactively convert complex external stimuli (e.g., light, electricity, humidity, magnetism, and pressure) into mechanical deformations, flexible actuators have fueled the continued development of soft robots in areas such as flexible grippers, energy harvesting, and wearable technology [1,2,3]. The gripper is the key component for the robots to directly interact with the external world. However, traditional mechanical grippers are prone to damage when gripping irregularly shaped or fragile items. Therefore, a large number of flexible grippers with non-destructive grasping and self-use grasping capabilities have emerged [4,5,6]. There are various driving methods for flexible grippers, mainly including rope driving [7], fluid driving [8], and intelligent material driving [9]. Intelligent materials have become a research hotspot due to their simple actuation method and excellent compliance [10,11,12]. As a kind of intelligent material, a dielectric elastomer (DE) can produce certain deformation under the excitation of an electric field. According to this principle, a DE can be developed into a dielectric elastomer actuator (DEA) [13,14,15]. A DEA is expected to be applied to the development of flexible grippers due to its low elastic modulus [16], large driving stroke [17], and fast response time [18]. However, the bendable DEA usually has a small deformation range and low output force when prepared as a flexible gripper, which limits the application of the gripper. Therefore, exploring the design and processing methods of bendable DEAs with a large deformation range and high output force has become an urgent problem to be solved.

Usually, a DEA is composed of DE film, a flexible electrode, and a passive layer [19]. An acrylic elastomer (VHB4910) has the advantages of high viscosity, low elastic modulus, and a high dielectric constant. The advantage of high viscosity makes it easy to combine with other materials, while the low elastic modulus and high dielectric constant result in significant electrical deformation at the same voltage [20,21,22]. Carbon grease has become the most common flexible electrode material due to its good conductivity and high flexibility [23]. Polydimethylsiloxane (PDMS) has excellent flexibility and stability, and is the most ideal material for the passive layer [24]. Pelrine et al. [25] applied pre-stretch on VHB4910 for the first time. When the pre-stretch of the circular film is 15% and the electric field strength is 55 V/Lm, the area strain is 40%. When the pre-strain is 300% and the electric field strength is 412 V/Lm, the area strain is 158%. And in the subsequent research, it was shown that the material had extremely high energy density, which opened the application of VHB4910 in the DEA [26]. Qiu et al. [27] showed that pre-stretch of the DE is required in the preparation of the DEA, because pre-stretch of the film can not only effectively reduce the driving voltage of the DE, but also avoid the failure of electromechanical instability to produce a large electro-induced deformation. Pre-stretch also helps to improve the breakdown strength of the DE. Studies have shown that when the pre-strain increases from 0 to 500%, the breakdown voltage increases from 18 V/Lm to 218 V/Lm [28]. However, the relationship between the pre-stretch ratio and DEA deformation degree has not been analyzed. Kofod et al. [29] proposed the minimum energy structure of the DEA, which is composed of a pre-stretch dielectric elastomer film and a flexible plastic frame. After the production is completed, the DEA presents a bending state, and the bending gradually flattens after energization. At present, the bendable DEA is usually the minimum energy structure. However, the use of plastic frames will still limit the overall deformation of the DEA, and may also cause structural failure due to plastic deformation of the frame. Araromi et al. [30] proposed a minimum energy structure of a curling dielectric elastomer, which is connected to a flexible frame by a pre-stretch dielectric elastomer actuator membrane. When the pre-stretch is released, the actuator finds a balance in the bending, and the bending angle can be changed by applying a voltage bias. This bending of the satellite-mounted gripping device causes the DEA’s maximum bending angle to change by more than 60°, but shows only a few microns of gripping force, which is enough to catch and hold objects in space, but not enough on Earth due to gravity [31]. Su et al. [32] prepared the improved DE by 3D printing, and on this basis, the rectangular DEA was prepared. The DEA can produce a bending deformation of 21° when the thickness of the elastomer is 60 microns, and the bending deformation increases with the increase in the film thickness. Shintake et al. [33] prepared the DEA of the interdigital electrode and the rectangular electrode, respectively. The results show that the DEA of the rectangular electrode has a larger deformation range and a higher output force than the DEA of the interdigital electrode when the thickness and size of the elastomer are the same. In addition, due to the increasingly complex tasks required to be performed by the new generation of intelligent robots, the integration of various practical functions on traditional actuators is an inevitable path for the future development of smart actuators. For example, Weng et al. [34] proposed polypyrrole@graphene-bacterial cellulose (PPy@G-BC) films to construct multi-responsive and bilayer actuators integrated with a multi-mode self-powered sensing function, which can perceive external temperature and humidity while moving. Zhou et al. [35] proposed a soft actuator with a self-powered sensing function based on the photo-thermal-electrical coupling mechanism, which can perceive materials based on the triboelectric effect. However, the integration of complex sensors imposes higher demands on the actuators’ deformation capabilities and driving forces. The previous studies have shown that VHB4910 is suitable for the preparation of the DEA, carbon grease can be used as a flexible electrode, and PDMS can be used as a passive layer to maintain good compliance of the DEA. However, the relationship between pre-stretch and DEA deformation needs to be clarified, and the design and preparation of a DEA with a large deformation range and high output force that can be applied to flexible gripper needs further research.

Based on the above analysis, this research explores the relationship between pre-stretch and DEA deformation degree, and further improves the deformation range and output force of DEAs through shape design, which expands the application scenario of the DEA in the flexible gripper. Firstly, the actuation principle of the DEA with bendable deformation was analyzed, and the structure and preparation process of DEAs with different shapes was designed. Secondly, the hyperelastic properties and relative dielectric constant of the DE were tested, and the bending deformation process of DEAs with different shapes was simulated by Finite Element Method (FEM) simulation. Thirdly, a rectangular DEA, trapezium DEA, and elliptical DEA were prepared, and the variation in the DEA bending angle and output force with increasing voltage was tested. Finally, a flexible gripper combined with a three-finger gripper framework and the elliptical DEA was developed. The balloons and paper cups grasping tests indicate that the prepared elliptical DEA can produce large bending deformation and high output force, and the flexible gripper composed of an elliptical DEA has good grasping ability.

## 2. Theories and Experiments

### 2.1. Actuator Principle and Structure Design

Dielectric elastomers (DEs) are a type of non-conductive and elastic material, which make them easy to deform after external electrical stimulation. According to this characteristic, a DE can be designed as an actuator, namely a dielectric elastomer actuator (DEA) driven by electricity. As shown in Figure 1, a DEA is similar to a sandwich structure, with a DE in the middle and conductive flexible electrodes covered on the upper and lower surfaces. When the DEA is in the initial state, the size of the DE is *L*_1_ × *L*_2_ × *L*_3_. When the voltage is applied to the flexible electrodes, the DEA will change to the drive state and the size of the DE will change to *l*_1_ × *l*_2_ × *l*_3_. Since the flexible electrodes at both ends will have opposite charges, the flexible electrodes will generate electrostatic forces that attract each other, causing the DE to be squeezed and generating Maxwell stress. The Maxwell stress compresses the thickness of the DE from *L*_3_ to *l*_3_. The decrease in thickness leads to an extension on the plane, and the area in the plane increases from *L*_1_ × *L*_2_ to *l*_1_ × *l*_2_. When the voltage is removed, the DE will return to the initial state. The magnitude of Maxwell stress is related to many factors and can be calculated by the following formula [36],(1)$$p=\varepsilon_{0}\varepsilon_{r}(\frac{V}{h}{)}^{2}$$
where p is the Maxwell stress, $\varepsilon_{0}$ and $\varepsilon_{r}$ are the vacuum dielectric constant and the relative dielectric constant of DE, V is the applied voltage, and h is the thickness of the DE between the two electrodes.

In previous research, when DEs were used to manufacture actuators, pre-stretch was often required [13]. Pre-stretch can enhance the deformation ability of the DE and avoid the occurrence of large electrical deformation due to mechanical and electrical instability failure [37,38,39]. Based on the above analysis, a DEA that can produce bending deformation was designed, as shown in Figure 2. The DEA was composed of a pre-stretch DE, a flexible electrode covering the upper and lower surfaces of the DE, and a thin passive layer and thick passive layer; and each layer was closely compacted. After being pre-stretched, tensile stress will be generated inside the DE, and both the thin passive layer and the thick passive layer will limit the shrinkage of the DE, but the limiting effect of the thin passive layer will be weaker than that of the thick layer. Therefore, the DEA will bend to the side of the thin passive layer. When the upper and lower flexible electrodes of the DEA are energized, Maxwell stress will be generated inside the DE to offset the original tensile stress, the DEA, and gradually restore the straight state.

### 2.2. Preparation of DEA

The preparation schematic diagram of the DEA is shown in Figure 3. Firstly, the DE was fixed on the stretching device, which can stretch the DE horizontally and vertically. After stretching the DE according to the required stretching conditions, it was fixed on the framework and removed from the stretching device. Secondly, the release liner was laser cut to obtain the required dimensions of rectangular, trapezium, and elliptical flexible electrodes. The obtained release liner mask was covered on both sides of the DE, and carbon grease was coated on the mask to obtain the desired shape of flexible electrodes. The shape and size of the two flexible electrodes are equal and overlap, and each flexible electrode is connected to a wire. The wires on the upper and lower electrodes are staggered and connected to the high voltage end and ground, respectively, when driving after completing the sample production. Finally, laser cutting was used to obtain the desired shapes of rectangular, trapezium, and elliptical thin and thick passive layers. The thin and thick passive layers were, respectively, covered on the upper and lower surfaces of the DE, and the DE was cut from the framework along the outer contour of the passive layer to obtain a DEA that can produce bending deformation.

To achieve significant deformation, the DE is typically required to have a high dielectric constant and a low elastic modulus. Common dielectric elastomer materials include acrylics, silicones, and polyurethanes [40]. Acrylic elastomers can produce large deformations and are easy to combine with other materials, but they usually exhibit hysteresis in response time [41]. Silicones have lower viscoelasticity, which makes their response time faster, but their lower dielectric constant requires a higher driving voltage [42]. Polyurethanes, although having a higher dielectric constant, have a higher elastic modulus, making it difficult to achieve large deformations [40]. Flexible electrodes are generally required to have good conductivity, high flexibility, and strong stability. Carbon grease, which meets the above requirements, also has the advantages of being inexpensive and highly reusable, making it an excellent choice for flexible electrodes. The passive layer material needs to have high flexibility and strong bonding ability, and it should be easy to fabricate. PDMS, with its simple and controllable fabrication process, is an ideal material for the passive layer.

The materials of each layer of the DEA were as follows: DE film adopted VHB4910 acrylic adhesive tape (produced by 3M company, Saint Paul, MN, USA), with a thickness of 1 mm, good ductility and viscosity, which is beneficial to the pre-stretch and adhesion of passive layer. The flexible electrode adopted 846 type carbon grease (produced by MG Chemical Company, Burlington, ON, Canada), which has good conductivity and structural compliance, and will not hinder the deformation of the DEA. The passive layer is made of DC184 silicone resin (produced by Dow Corning company, New York, NY, USA), and the required thickness of the passive layer is obtained by coating.

### 2.3. Performance Testing of DE

To achieve accurate prediction of the DEA deformation process, the mechanical and dielectric tests of the DE were carried out, respectively. When DE materials undergo their own deformation several times, it is difficult for the relationship between stress and strain to be described by the traditional linear elastic model. Therefore, the hyperelastic theory that can accurately describe large deformation can be used. For DE, the Yeoh model can be used to describe hyperelastic deformation [43]; it can be described by the following formula,(2)$$W_{S}=C_{10}(I_{1}-3)+C_{20}(I_{1}-3{)}^{2}+C_{30}(I_{1}-3{)}^{3}$$
where $W_{S}$ is the strain energy of DE, $C_{10}$, $C_{20}$, $C_{30}$ are the characteristic parameters of the Yeoh model, and $I_{1}$ is the first-order variable in the left Cauchy–Green deformation tensor.

The relationship between stress and strain can be obtained by further deducing the Yeoh model,(3)$$\sigma =2\left[\frac{\varepsilon^{3}+3\varepsilon^{2}+3\varepsilon }{{\left(1+\varepsilon \right)}^{2}}\right]\left[C_{10}+2C_{20}\frac{\varepsilon^{3}+3\varepsilon^{2}}{1+\varepsilon }+3C_{30}{\left(\frac{\varepsilon^{3}+3\varepsilon^{2}}{1+\varepsilon }\right)}^{2}\right]$$
where σ is stress, and ε is strain.

In order to obtain the parameters of the Yeoh model, it is necessary to carry out a uniaxial tensile test on the DE. As shown in Figure 4, according to the existing elastic body tensile test standard ASTM D412 [44], dumbbell-shaped tensile test samples were prepared and uniaxial tensile tests were carried out. The parameters of the Yeoh model can be obtained by fitting the obtained stress–strain curve.

Due to the frequent need for pre-stretching of the DEA during use, the dielectric properties of the DE may vary at different pre-stretching rates. Therefore, it is necessary to test the relative dielectric constant of the DE at different pre-stretching ratios. The DE with different pre-stretch ratios was obtained by the above stretching device, as shown in Figure 5. Because the DE (VHB4910) had strong viscosity, it was not conducive to the replacement of the sample during the test process. Therefore, the surface of the sample was covered with a conductive tin paper film, which has good conductivity and will not affect the test results. Through the analysis of the deformation principle and basic characteristics of the DE, it can be seen that the DE can be considered as a parallel plate capacitor. The relative dielectric constant can be tested by the capacitance method, and the relative dielectric constant $\varepsilon_{r}$ can be calculated by the following formula,(4)$$\varepsilon_{r}=\frac{C\cdot z}{A\cdot \varepsilon_{0}}$$
where C is the capacitance of the parallel plate capacitor, A is the area of the parallel plate capacitor plate, z is the thickness of the dielectric between the two plates of the parallel plate capacitor, and $\varepsilon_{0}$ is the dielectric constant in a vacuum (8.854 × 10^−12^ F/m).

### 2.4. Characterizations

A three-dimensional model of the DEA was established using the finite element analysis software ABAQUS 2020 (Dassault Systèmes SOLIDWORKS Corp, Waltham, MA, USA) and the deformation process after power on was simulated. The simulation used a three-dimensional hexahedral mesh with hybrid elements (C3D8H). The relationship between the elongation of the DEA and voltage under different pre-stretch ratios was obtained by MATLAB 2020 (Mathworks Inc., Natick, MA, USA). The release liner mask and silicone resin passive layer with precise shape were prepared by a laser cutting machine (ZGD-4060, Shandong Zhongguang Optoelectronic Technology Co., Ltd., Zibo, China). The high-voltage electricity to drive DEA deformation was provided by DC high-voltage power (DW-P503-1ACD1, Dongwen High Voltage Power Supply Co., Ltd., Tianjin, China). The mechanical properties of the DE were tested by an electronic universal material testing machine (ZQ-990, Dongguan Zhizhi Precision Instrument Co., Ltd., Dongguan, China). The dielectric constant of the DE under different pre-stretch ratios was tested by a Dielectric constant tester (ZJD-A, Beijing AVIC Times Instrument Equipment Co., Ltd., Beijing, China). The output force of the DEA was tested by a miniature pull pressure sensor (ZNLBS, Bengbu Zhongnuo Sensor Co., Ltd., Bengbu, China).

## 3. Result and Discussion

### 3.1. The Relationship Between Pre-Stretch Ratio and DEA Deformation Degree

Pre-stretch has a significant effect on the performance degree of the DEA. In order to further clarify the relationship between pre-stretch and DEA deformation, the manufacturing of a square DEA under uniaxial stretching was completed. The initial DEA size was 100 mm × 100 mm × 1 mm, the covering electrode size was 30 mm × 30 mm, and the uniaxial stretching ratios (λ) were 1.25, 1.5, 1.75, 2, 2.25, 2.5, 2.75, and 3. The high-voltage DC power was used to provide the driving voltage, and the camera was used to capture the change process of the DEA during the application of the voltage. The same pre-stretch test ensured the same shooting conditions, and the image was processed to obtain the elongation of the electrode area.

The uniaxially stretched square DEA is slowly loaded from 0 V, so that it is completely strained during the power supply cycle until the voltage is too large to break it down. The deformation process is shown in Figure 6a, the vertical direction is the stretching direction, and the transverse direction is the unstretched direction. The DE materials have a large Poisson’s ratio and can be considered incompressible materials. Therefore, when the vertical direction is stretched, the transverse contraction of the DE is hindered, and the transverse direction is also passively stretched. When the voltage gradually increases, the DEA covering the electrode area gradually begins to stretch, and the expansion phenomenon occurs in both transverse and vertical directions. The pre-stretch ratio in the stretching direction is greater than in the unstretched direction. Therefore, during the application of voltage, the elongation in the stretching direction is greater than in the unstretched direction. Macroscopically, as the applied voltage increases, wrinkles appear in the vertical direction of the dielectric elastomer. However, as the vertical stretching ratio gradually increases, the passive stretching ratio in the horizontal direction also gradually increases, so the wrinkles gradually disappear.

As shown in Figure 6b, the relationship between transverse elongation of the square DEA and voltage, and ‘*’ is the breakdown point. In the transverse direction, with the gradual increase in the applied voltage, the transverse side length of the square DEA increases gradually, and with the increase in the pre-stretch ratio, the elongation gradually increases. When the uniaxial pre-stretch ratio is 1.25, the minimum elongation is 2.69 mm. When the uniaxial pre-stretch ratio is 3, the maximum elongation is 3.77 mm. As shown in Figure 6c, in the vertical direction, with the gradual increase in the applied voltage, the vertical side length of the square DEA increases gradually, and with the increase in the pre-stretch ratio, the elongation gradually decreases. When the uniaxial pre-stretch ratio is 1.25, the maximum elongation is 6.10 mm. When the uniaxial pre-stretch ratio is 3, the minimum elongation is 4.19 mm. In general, as the pre-stretch ratio increases, the breakdown voltage gradually decreases. This is because as the pre-stretch ratio increases, the thickness of the film decreases and the ratio of elongation change with voltage also increases. However, the transverse elongation is always smaller than the vertical elongation, which also indicates that the passive stretching ratio is always smaller than the active stretching ratio.

The parameters of the square DEA under biaxial stretching are the same as those under uniaxial stretching. The biaxial stretching square DEA is prepared, and the biaxial stretching ratios (λ) are 1.25, 1.5, 1.75, 2, 2.25, 2.5, 2.75, and 3. The biaxial stretching square DEA is slowly loaded from 0 V, so that it is completely strained during the power supply cycle until the voltage is too high to break it down. As shown in Figure 7a, when the external voltage is gradually increased, the DEA covering the electrode area gradually begins to stretch, and the expansion phenomenon occurs in both the transverse tensile direction and the vertical tensile direction. Different from uniaxial tension, due to the same transverse and vertical pre-stretch ratios, the transverse and vertical expansion degrees are basically the same during the application of voltage, so no wrinkles are generated. However, due to the transverse side length of the square DEA being pasted with conductive tape, which limits the further expansion of the transverse, the transverse elongation will be slightly smaller than the vertical elongation, and the closer to the conductive tape, the smaller the area expansion.

As shown in Figure 7b, the transverse elongation of the square DEA under biaxial stretching is limited due to the effect of conductive wire in the transverse stretching direction. With the increase in the applied voltage, the side length of the square DEA in the transverse stretching direction increases gradually, and the elongation increases with the increase in the pre-stretch rate. When the biaxial pre-stretch ratio is 1.25, the minimum elongation is 4.42 mm. When the biaxial pre-stretch ratio is 3, the maximum elongation is 8.64 mm. As shown in Figure 7c, with the gradual increase in the applied voltage, the side length of the square DEA in the vertical stretching direction gradually increases, and with the increase in the pre-stretch ratio, the elongation gradually increases. When the biaxial pre-stretch ratio is 1.25, the minimum elongation is 7.11 mm. When the biaxial pre-stretch ratio is 3, the maximum elongation is 12.12 mm. Due to the limitation of conductive wire, the elongation in the transverse stretching direction is always smaller than that in the vertical stretching direction.

The DEA designed in this research is mainly to produce bending deformation, so it can be achieved only by uniaxial stretching. In order to make a DEA have greater deformability, the uniaxial stretching ratio should be as small as possible, but a smaller stretching ratio is difficult to produce a larger initial bending deformation, and the specific uniaxial stretching ratio needs to be further analyzed and determined.

### 3.2. Shape Design of DEA

As shown in Figure 8a, the hyperelastic properties of the DE are described by the Yeoh model. The parameters of the Yeoh model can be obtained by fitting the uniaxial tensile stress–strain curve, and $C_{10}$, $C_{20}$, and $C_{30}$ are 4.36 × 10^−3^ Mpa, 4.16 × 10^−4^ Mpa, and −1.33 × 10^−5^ Mpa, respectively. As shown in Figure 8b, the relative dielectric constant of the DE material at different pre-stretch ratios can be measured by the instrument. With the increase in the pre-stretch ratio, the relative dielectric constant decreases gradually. When the uniaxial tensile ratio is 1.75, the relative dielectric constant is 3.96.

The bending deformation range of the DEA is related to the pre-stretch ratio of the DE, the hyperelastic properties of the DE, the parameters of the passive layer, and its own shape. Therefore, it is necessary to build a finite element model for systematic investigation. The finite element analysis of the DEA bending deformation process is divided into three steps. The first step is the pre-stretch of the DE material. One end of the DE material is completely fixed, and the other end is stretched to the required pre-stretch ratio. The second step is to combine the pre-stretch DE with PDMS to form the DEA, and the surface of the DE contact with PDMS is set to be bound. The third step is to apply electrostatic pressure to the DEA, and apply electrostatic pressure on the upper and lower surfaces of the DE, and the magnitude of electrostatic pressure can be calculated by Formula (1). The mesh type of the DE is C3D8H, and the mesh size is 0.5 mm. The material properties of the DE are input according to the Yeoh hyperelastic model. The mesh type of PDMS is C3D8R, and the mesh size is 0.5 mm. The material properties of PDMS are set to be linearly elastic, the elastic modulus is set to 2 Mpa, and the Poisson’s ratio is 0.45 [45]. In the simulation model, the DEA area is set to 1000 mm^2^, and the DEA size is simplified and optimized based on the principle of controlling variables. The size of the rectangular DEA is 20 mm × 50 mm, the size of the trapezium DEA is 25 mm × 15 mm × 50 mm, and the size of the elliptical DEA is 50 mm × 12.74 mm. In addition, in order to obtain a larger or smaller DEA, the size can be increased or decreased year-on-year.

A DEA with bending deformation needs a DE after uniaxial stretching. In the process of exploring the relationship between the uniaxial pre-stretch ratio and DEA deformation, the larger the pre-stretch ratio, the smaller the deformation of the DEA. Therefore, a smaller pre-stretch ratio should be selected as much as possible in the preparation of a DEA with bending deformation. However, as the pre-stretch ratio decreases, the initial bending angle of the DEA will also decrease. Therefore, a smaller stretching rate should be selected as much as possible to ensure a sufficient initial bending angle. Figure 9 shows the initial bending angle of the DEA under different pre-stretch ratios. When the pre-stretch ratios are 1.25, 1.5, and 1.75, the initial bending angles of the DEA are 45°, 30°, and 21°. In order to meet the conditions of the flexible gripper, the uniaxial pre-stretch ratio of the DEA is set to 1.75.

Figure 10 shows the displacement nephogram of the rectangular DEA, trapezium DEA and elliptical DEA at 0, 10, and 20 kV. The results show that the rectangular DEA, elliptical DEA, and rectangular DEA all change from the bending state to the flat state after the voltage is applied. However, the initial deformation and the deformation range after applying voltage are different. The initial deformation and deformation range of the elliptical DEA are the largest, followed by the trapezium DEA, and the rectangular DEA is the smallest.

### 3.3. Performance Test of DEA

In order to verify the influence of the DEA shape on bending deformation, the rectangular DEA, trapezium DEA, and elliptical DEA with the same parameters as those in the finite element analysis model were prepared. The voltage was applied from 0 V until the breakdown, the bending deformation process was recorded, and the bending angle and the output force were tested. Figure 11 shows the bending deformation states of the rectangular DEA, the trapezium DEA, and the elliptical DEA at voltages of 0, 8, 16, and 24 kV. The DEA breaks down when the voltage exceeds 24 kV. The reason why the breakdown voltage is higher than that of the square DEA uniaxial test is that the passive layer covers the upper and lower surfaces of the DEA, which limits the expansion of the DE material. The thickness does not decrease, so the breakdown voltage will increase. The DEA deformation process of three different collisions is consistent with the FEM results. The initial bending angle and deformation range of the elliptical DEA are the largest, and the initial bending angle and deformation range of the rectangular DEA are the smallest.

Figure 12a shows the change in the bending angle of the rectangular DEA, trapezium DEA, and elliptical DEA after applying voltage. When the voltage is 0 V, the position is the initial angle of the DEA, which is recorded as the zero point. After the voltage is applied, the angle of the DEA minus the initial angle is the current bending angle. With the increase in voltage, the bending angle of the DEA with three shapes increases gradually, and the bending range of the elliptical DEA is the largest. The maximum bending angles of the rectangular DEA, the trapezium DEA, and the elliptical DEA are 25°, 32°, and 40°, respectively. Figure 12b shows the changes in the output force of the rectangular DEA, the trapezium DEA, and the elliptical DEA after applying voltage. During the test, the angle of the sensor was adjusted to stay the same as the initial angle of the DEA, and the sensor could gradually detect the output force of the DEA after applying voltage. With the increase in voltage, the output force of the three shapes of DEA increases gradually, and the output force range of the elliptical DEA is the largest. The maximum output forces of the rectangular DEA, trapezium DEA, and elliptical DEA are 24.3 mN, 29.0 mN, and 37.2 mN, respectively. The bending deformation degree and output force of the DEA have been improved compared with previous reports [31,33].

Figure 13 shows the difference between the DEA bending deformation in the simulation model and the experiment. The DEA bending deformation of the simulation model is greater than that of the DEA in the experiment. There are three main reasons for this phenomenon. Firstly, PDMS undergoes certain hardening at the edges after laser cutting. Secondly, there are some errors in the hyperelastic fitting of VHB4910. Finally, there may also be some errors in the testing process of the dielectric constant.

### 3.4. Grasp Test of the Designed Flexible Gripper

The elliptical DEA can produce large bending deformation and can have high output force. Based on the elliptical DEA, a three-finger flexible gripper was prepared. Figure 14a shows the frame size of the three-fingered flexible gripper. Figure 14b shows the three-dimensional model of the three-fingered flexible gripper. A single gripper was fixed on a rectangular plane, three rectangular planes were cut out of a circle with a radius of 20 mm, and the angle between two adjacent rectangular planes was 120°. As shown in Figure 14c, three elliptical DEAs were prepared and fixed on the three-finger gripper frame. The positive electrodes of the three grippers were connected to the upper surface of the gripper frame, and the negative electrodes were connected to the lower surface of the gripper frame.

As shown in Figure 14d–f, the three-finger flexible gripper was used to test the grasping of balloons. When the voltage was 0, the three-finger flexible gripper was adjusted to the appropriate position, and then the voltage was gradually increased to 20 kV. The three-finger flexible gripper was closely attached to the balloon, and then the voltage was kept unchanged to successfully grab the balloon. As shown in Figure 14g–i, the three-fingered flexible gripper was used to test the grasping of paper cups. The initial weight of the paper cup was 5.82 g. In order to ensure successful grabbing, when the gripper was in contact with the surface of the paper cup, the voltage was increased to 24 kV, and the weight of the paper cup was reduced to 4.34 g. After that, the voltage was kept constant and the paper cup was successfully grabbed.

## 4. Conclusions

This research improved the bending deformation degree and the output force of the DEA through an appropriate pre-stretch ratio and shape design, which provides a technical reference for the application of intelligent flexible actuators integrating complex sensing functions in the future. The relationship between the pre-stretch ratio and DEA deformation degree was tested by experiments, the bending deformation process of DEAs with different shapes under pressure was simulated by FEM simulation based on the performance test results of the dielectric elastomer, DEAs with different shapes were prepared through laser cutting, and the relationship between the voltage and the bending angle and the output force of the DEAs was measured.
(1)Under uniaxial stretching, the deformation of the DEA in the stretching direction gradually increases and decreases in the unstretched direction with the increase in the pre-stretch ratio. Under biaxial stretching, DEA deformation increases with the increase in the pre-stretch ratio.(2)The shape of the DEA has a certain influence on the bending deformation range under the same conditions, and the elliptical DEA has a larger bending deformation range and higher output force compared with the rectangular DEA and the trapezium DEA.(3)The bending angle curve and output force curve after applying voltage to the elliptical DEA show that the elliptical DEA can produce a maximum bending deformation of 40° and an output force of 37.2 mN at a voltage of 24 kV.(4)Looking ahead, flexible sensors with high sensitivity and a wide linear range will be arrayed and integrated into the flexible gripper proposed in this research. This integration is expected to show great application scenarios in the fields of smart agriculture, human–computer interaction, and soft robots.

## Figures and Tables

**Figure 1 micromachines-16-00107-f001:**
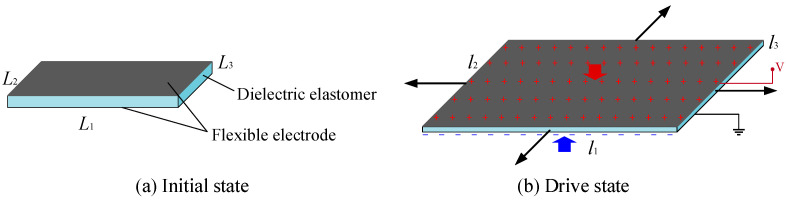
The driving principle of DEAs.

**Figure 2 micromachines-16-00107-f002:**
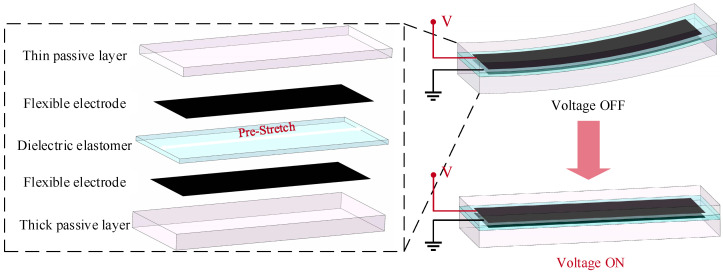
The structure and deformation process of DEAs.

**Figure 3 micromachines-16-00107-f003:**
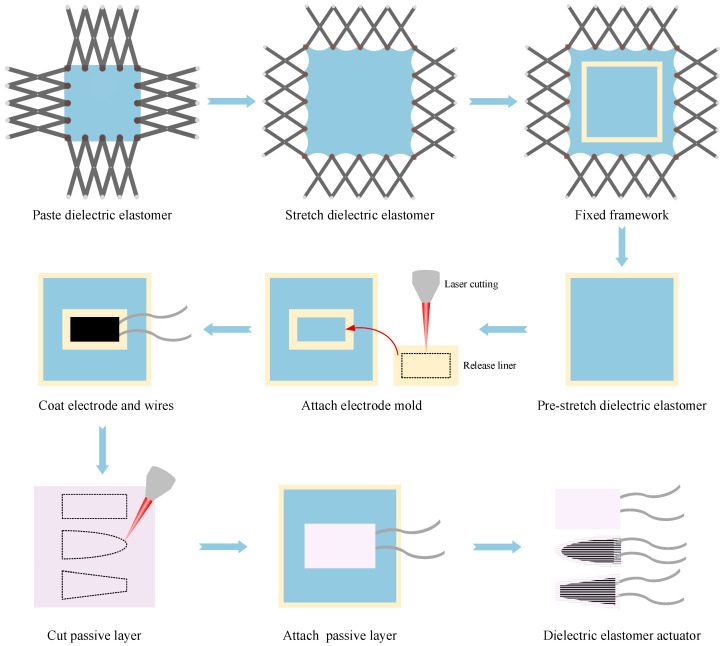
The preparation schematic diagram of the DEA.

**Figure 4 micromachines-16-00107-f004:**
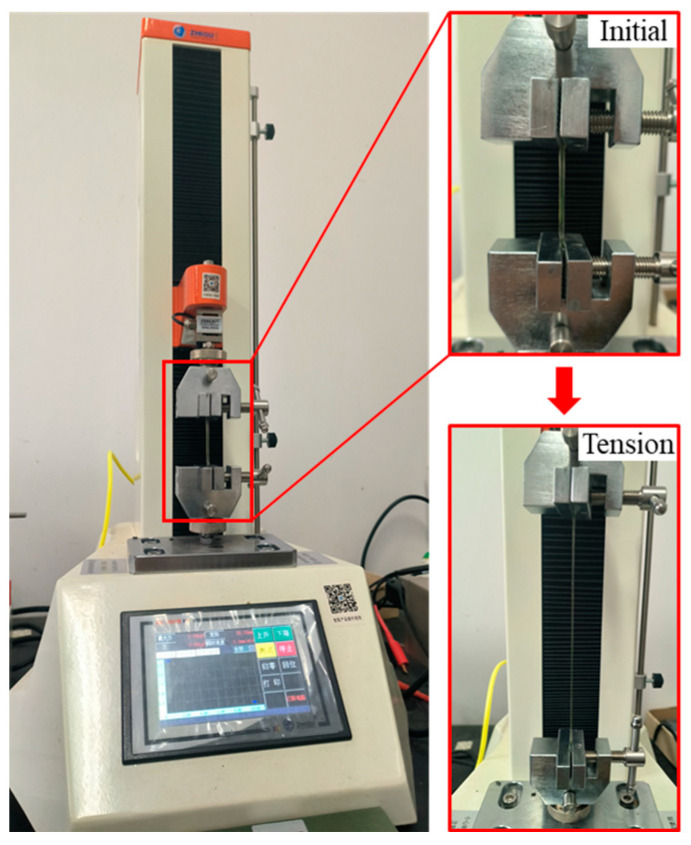
Hyperelastic tensile test.

**Figure 5 micromachines-16-00107-f005:**
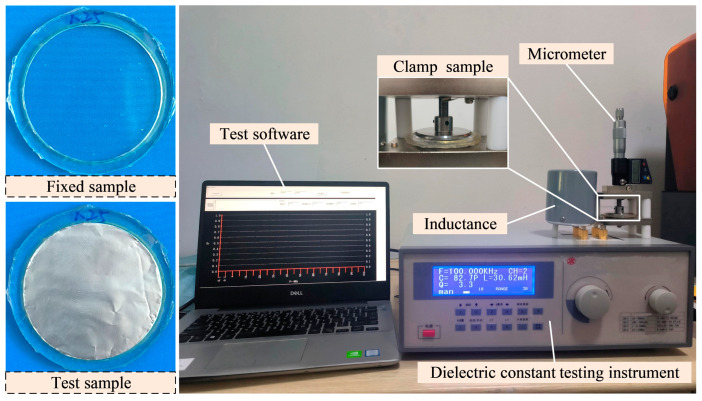
Relative dielectric constant test.

**Figure 6 micromachines-16-00107-f006:**
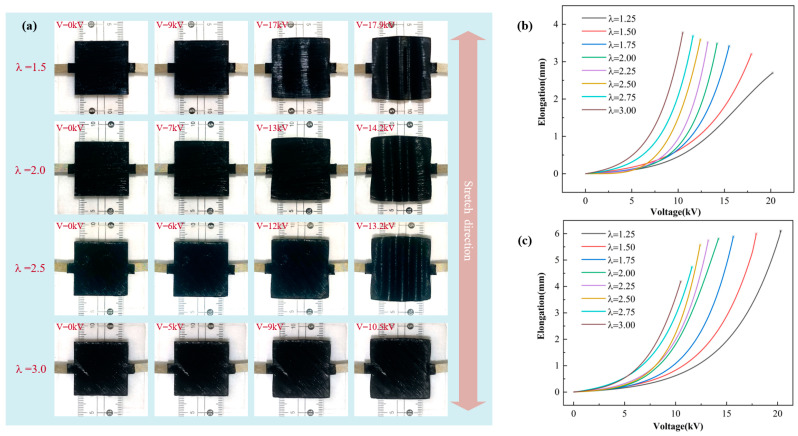
The test results under uniaxial stretching, (**a**) the deformation process of square DEA, (**b**) the relationship between transverse elongation and voltage, and (**c**) the relationship between vertical elongation and voltage.

**Figure 7 micromachines-16-00107-f007:**
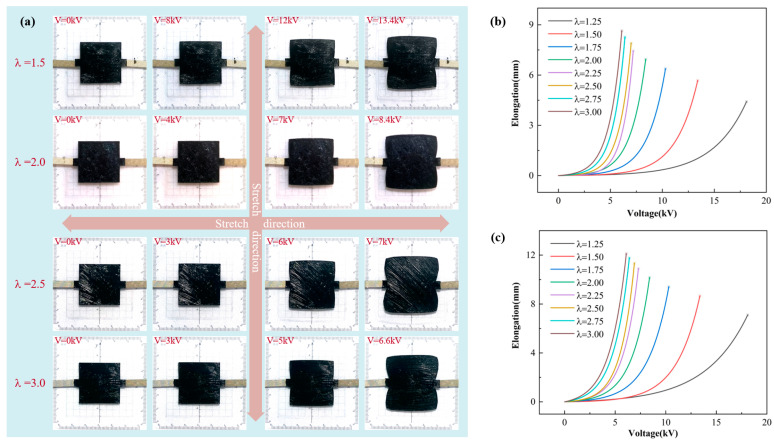
The test results under biaxial stretching, (**a**) the deformation process of square DEA, (**b**) the relationship between transverse elongation and voltage, and (**c**) the relationship between vertical elongation and voltage.

**Figure 8 micromachines-16-00107-f008:**
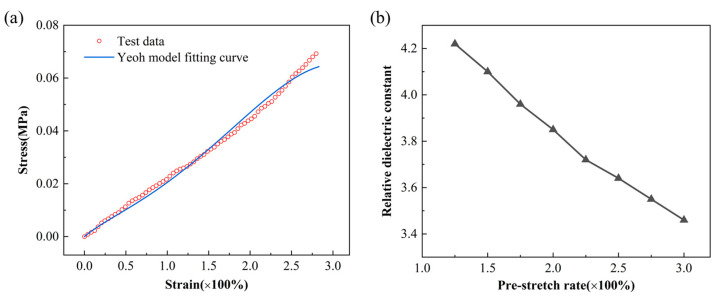
Material test results, (**a**) hyperelastic test results, and (**b**) relative dielectric constant test results.

**Figure 9 micromachines-16-00107-f009:**
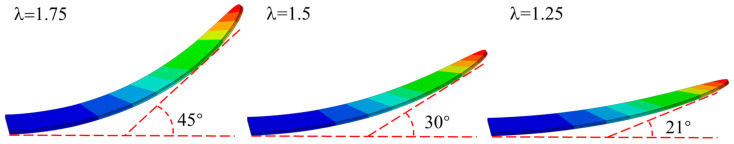
The initial bending angle of DEA when the pre-stretch ratio is 1.25, 1.50, and 1.75.

**Figure 10 micromachines-16-00107-f010:**
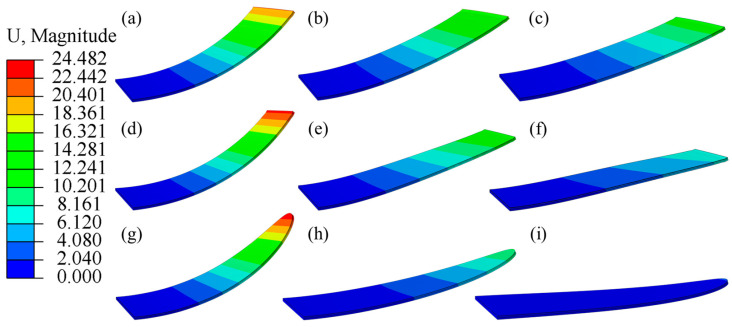
The displacement nephograms of DEAs with different shapes at voltages of 0 kV, 10 kV, and 20 kV; (**a**–**c**) the deformation process of rectangular DEA, (**d**–**f**) the deformation process of trapezium DEA, and (**g**–**i**) the deformation process of elliptical DEA.

**Figure 11 micromachines-16-00107-f011:**
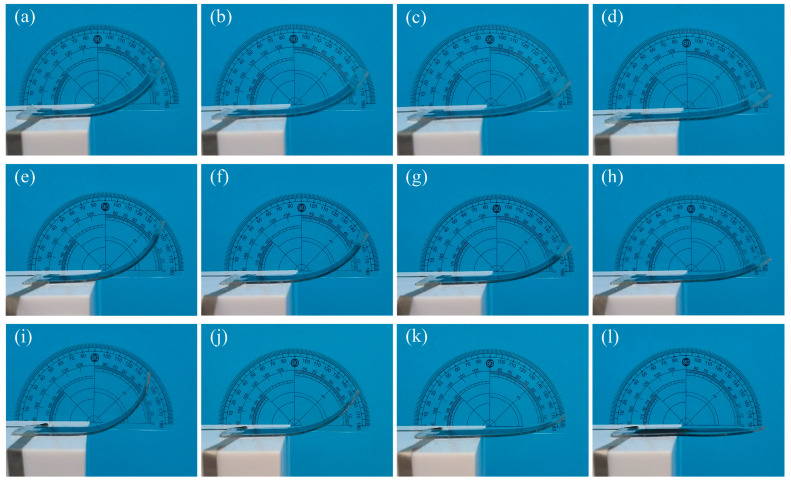
The bending state of DEAs with different shapes under voltages of 0 kV, 8 kV, 16 kV, and 24 kV; (**a**–**d**) the deformation process of rectangular DEA, (**e**–**h**) the deformation process of trapezium DEA, and (**i**–**l**) the deformation process of elliptical DEA.

**Figure 12 micromachines-16-00107-f012:**
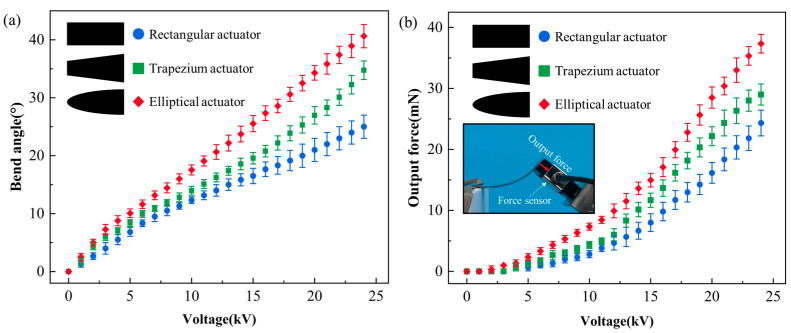
The performance test of DEA; (**a**) the change in bending angle with voltage, and (**b**) the change in output force with voltage.

**Figure 13 micromachines-16-00107-f013:**
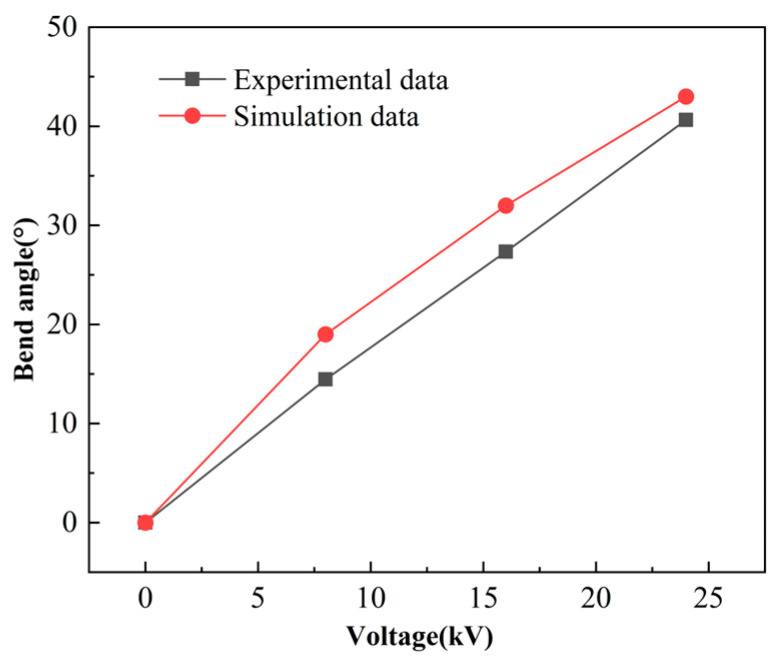
The difference between simulation data and experimental data.

**Figure 14 micromachines-16-00107-f014:**
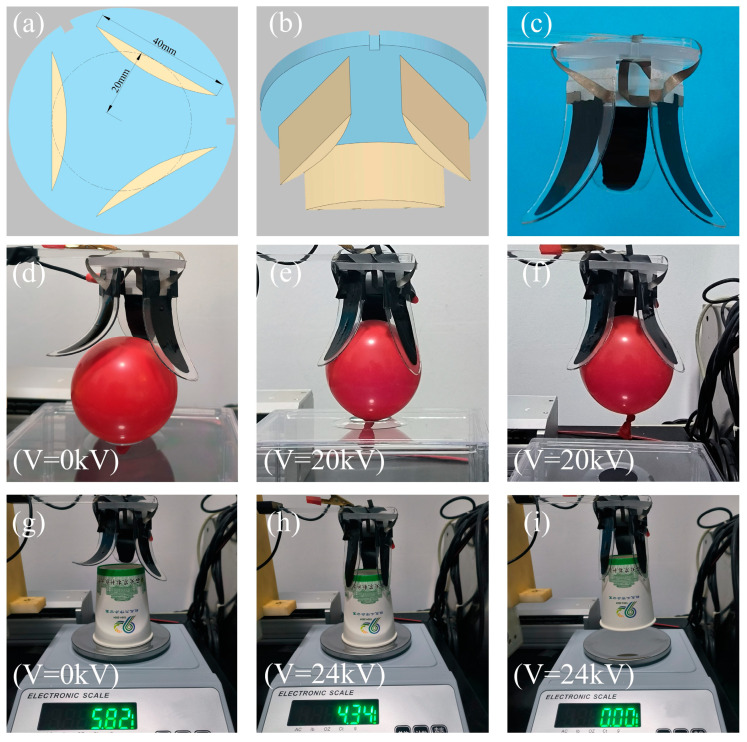
The grabbing test of three-fingered gripper, (**a**) the frame size of the gripper, (**b**) the three-dimensional model of the gripper, (**c**) three-finger gripper physical map, (**d**–**f**) test of grasping balloons, and (**g**–**i**) test of grasping paper cups.

## Data Availability

The original contributions presented in the study are included in the article, further inquiries can be directed to the corresponding author.

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
