# Peer review of "Design of Dielectric Elastomer Actuator and Its Application in Flexible Gripper"

_micromachines, 2025, doi:10.3390/mi16010107_

Round 1
Reviewer 1 Report
Comments and Suggestions for Authors
General Comments:
The manuscript presents a comprehensive study on the design and application of a dielectric elastomer actuator (DEA) for flexible grippers. The authors have successfully demonstrated the relationship between pre-stretch ratio and DEA deformation degree, simulated the bending deformation process using Finite Element Method (FEM), and tested the performance of DEAs with different shapes. The research is well-structured, and the results are promising for the development of flexible grippers with non-destructive grasping capabilities. However, the following questions and suggestions are provided to enhance the quality and depth of the manuscript.
1. The manuscript mentions the use of VHB4910 acrylic elastomer, carbon grease, and PDMS for the DEA. Could the authors elaborate on the material selection process and any potential trade-offs between the mechanical properties and electrical performance of these materials? Additionally, have any alternative materials been considered, and if so, what were the reasons for their exclusion?
2. The study investigates the effect of pre-stretch ratio on DEA deformation. What criteria were used to determine the optimal pre-stretch ratio for the elliptical DEA, and how does this ratio influence the long-term durability and reliability of the actuator? Are there any concerns regarding material fatigue due to repeated stretching and relaxation cycles?
3. The FEM simulation results are used to validate the experimental findings. Could the authors provide more details on the simulation parameters, such as the mesh size, boundary conditions, and material properties used in the FEM models? Additionally, how does the simulation accuracy compare with experimental data, and what are the sources of any discrepancies observed?
4. The manuscript highlights the influence of DEA shape on bending deformation range. What design principles or mathematical models were used to optimize the shape of the elliptical DEA for maximum bending deformation? Furthermore, could the authors discuss the scalability of the DEA design for larger or smaller grippers and the implications for performance?
5. The paper demonstrates the grasping capabilities of the elliptical DEA-based flexible gripper with balloons and paper cups. What are the limitations of the current design in terms of grasping a variety of objects with different weights, shapes, and surface properties? Additionally, how does the gripper's performance compare with existing solutions in terms of grasping force, precision, and adaptability? Also, some refs. (10.1016/j.nanoen.2024.110552, 10.1002/EXP.20210112, 10.1002/advs.202309846) about actuators integrated with e-skins are suggested to discuss or introduce in the manuscript.
Reviewer 2 Report
Comments and Suggestions for Authors
The subject of the paper is of great interest in developing original soft actuators for soft fingers / grippers and soft robotics.
Your paper represents useful work in this field. It seems that you have had other contributions in this field and in related domains and have published previous papers.
Please consider the following suggestions:
- please specify the original contributions of this paper, in relation to your other previous research and papers;
- please highlight some potential applications of the proposed system;
- please extend some of yours conclusions and add comments regarding your future work in this area;
- the writing style of the reference list is not the one provided in the MDPI Reference List and Citations Style Guide (https://www.mdpi.com/authors/references); please check if is necessary to change the style of the References list;
- minor English corrections are required.
Author Response
Author response:On behalf of my co-authors, we appreciate the reviewer very much for your positive and constructive comments and suggestions on our manuscript. Those comments are all valuable and very helpful for revising and improving our paper, as well as the important guiding significance to our research. Based on the reviewer’s comments, we improved and modified our research.
Response to Reviewer#2:
Comments 1: Please specify the original contributions of this paper, in relation to your other previous research and papers.
Response 1: Thank you for your valuable feedback and for giving us the opportunity to clarify the original contributions of this paper in relation to our previous work. This paper presents a new research direction, the design and fabrication methods for dielectric elastomer actuator (DEA) with large deformations and high output forces. In our previous work, we focused on the design and fabrication of flexible sensors with high sensitivity and broad linear range. In the future, we hope to array these sensors in combination with flexible actuators.This integration holds promise for developing a new generation of intelligent flexible gripper. However, current DEA are limited by their small deformation range and low output force, which restricts their further application. In this research, we investigate the relationship between pre-stretch and the maximum deformation of DEA. Furthermore, by altering the shape of the actuators, we have significantly increased the bending deformation range and output force. These findings provide a technical reference for the future application of intelligent flexible gripper.
Action 1: The following content was added to the manuscript.
Abstract
Dielectric elastomer actuator (DEA) are difficult to apply to flexible grippers due to their small deformation range and low output force. Hence, a DEA with a large bending deformation range and output force was designed, and corresponding flexible gripper was developed to realize the func-tion of grasping objects of different shapes.
Section 1
Based on the above analysis, this study explores the relationship between pre-stretch and DEA deformation degree, and further improves the deformation range and output force of DEA through shape design, which expands the application scenario of DEA in flexible gripper.
Comments 2: Please highlight some potential applications of the proposed system.
Response 2: Thank you for your insightful comment regarding the potential applications of the proposed system. The authors are pleased to elaborate on the various practical scenarios where our system can be effectively utilized and to include these applications in the manuscript.
Action 2: The following content was added to the manuscript.
Section 1
Benefiting from the capability to proactively convert complex external stimuli (e.g., light, electricity, humidity, magnetism, and pressure) into mechanical deformations, flexible actuators have fueled the continued development of soft robots in areas such as flexible grippers, energy harvesting, and wearable technology [1-3].
In addition, due to the increasingly complex tasks required to be performed by the new generation of intelligent robots, the integration of various practical functions on traditional actuators is an inevitable path for the future development of smart actuators. For example, Weng et al. [34] proposed a polypyrrole@graphene-bacterial cellulose (PPy@G-BC) films to construct multi-responsive and bilayer actuators integrated with multi-mode self-powered sensing function, which can perceive external temperature and humidity while moving. Zhou et al. [35] proposed a soft actuator with self-powered sensing function based on the photo-thermal-electrical coupling mechanism, which can perceive materials based on the triboelectric effect. However, the integration of complex sensors imposes higher demands on the actuators' deformation capabilities and driving forces.
References
[1]Rus, D; Tolley, M. T. Design, fabrication and control of soft robots. Nature. 2015, 521, 467-475. https://doi.org/10.1038/nature14543
[2]Park, J.; Lee, Y.; Cho, S.; Choe, A.; Yeom, J.; Ro, Y. G.; Ko, H. Soft Sensors and Actuators for Wearable Human–Machine Interfaces. Chem. Rev. 2024, 124, 1464-1534. https://doi.org/10.1021/acs.chemrev.3c00356
[3]Chen, Y.; Gao, Z.; Zhang, F.; Wen, Z.; Sun, X. Recent progress in self‐powered multifunctional e‐skin for advanced applications. Exploration. 2022, 2, 20210112. https://doi.org/10.1002/EXP.20210112
[34]Weng, M.; Zhou, J.; Zhou, P.; Shang, R.; You, M.; Shen, G.; Chen, H. Multi‐Functional Actuators Made with Biomass‐Based Graphene‐Polymer Films for Intelligent Gesture Recognition and Multi‐Mode Self‐Powered Sensing. Adv. Sci. 2024, 11, 2309846. https://doi.org/10.1002/advs.202309846
[35]Zhou, J.; Chen, H.; Wu, Z.; Zhou, P.; You, M.; Zheng, C.; Weng, M. 2D Ti3C2Tx MXene-based light-driven actuator with integrated structure for self-powered multi-modal intelligent perception assisted by neural network. Nano Energy. 2025. 134, 110552. https://doi.org/10.1016/j.nanoen.2024.110552
Comments 3: Please extend some of yours conclusions and add comments regarding your future work in this area.
Response 3: Thank you for your valuable feedback and for giving us the opportunity to further elaborate on our conclusions and future work in this area. The authors have added a prospect for future work in the conclusion.
Action 3: The following content was added to the manuscript.
Section 4
This research improved the bending deformation degree and the output force of DEA through appropriate pre-stretch ratio and shape design, which provides a technical reference for the application of intelligent flexible actuators integrating complex sensing functions in the future.
(4) Looking ahead, flexible sensors with high sensitivity and wide linear range will be arrayed and integrated into the flexible gripper proposed in this research. This integration is expected to show great application scenarios in the fields of smart agriculture, human-computer interaction, and soft robots.
Comments 4: The writing style of the reference list is not the one provided in the MDPI Reference List and Citations Style Guide (https://www.mdpi.com/authors/references); please check if is necessary to change the style of the References list.
Response 4: The authors are very grateful to the reviewers for pointing out the issue of the reference format. The authors have revised all references according to the style provided in the MDPI Reference List and Citations Style Guide.
Comments 5: Minor English corrections are required.
Response 5: The authors are very grateful to the reviewers for the revision comment. The authors have completed the language modifications for the entire text.